# Ivermectin Binds to the Allosteric Site (Site 2) and Inhibits Allosteric Integrin Activation by TNF and Other Pro-Inflammatory Cytokines

**DOI:** 10.3390/ijms26178655

**Published:** 2025-09-05

**Authors:** Yoko K. Takada, Yoshikazu Takada

**Affiliations:** 1Department of Dermatology, University of California School of Medicine, Research III Suite 3300, 4645 Second Ave., Sacramento, CA 95817, USA; yoktakada@ucdavis.edu; 2Department of Biochemistry and Molecular Medicine, University of California School of Medicine, Research III Suite 3300, 4645 Second Ave., Sacramento, CA 95817, USA; 3VA Northern California Health Care System, 150 Muir Road, Martinez, CA 94553, USA

**Keywords:** ivermectin, TNF, cytokine, integrin, allosteric integrin activation, allosteric site (site 2)

## Abstract

Ivermectin (IVM), a broad-spectrum anthelmintic agent, has anti-inflammatory properties, and affects cellular and humoral immune responses. We recently showed that multiple pro-inflammatory cytokines (e.g., FGF2, CCL5, CD40L) bind to the allosteric site (site 2) of integrins and activate them. 25-Hydroxycholesterol, a pro-inflammatory lipid mediator, is known to bind to site 2 and induce integrin activation and inflammatory signals (e.g., IL-6 and TNF secretion), suggesting that site 2 is critically involved in inflammation. We showed that two anti-inflammatory cytokines (FGF1 and NRG1) bind to site 2 and inhibit integrin activation by inflammatory cytokines. We hypothesized that ivermectin binds to site 2 and inhibits inflammatory signaling by pro-inflammatory cytokines. A docking simulation predicts that ivermectin binds to site 2. Ivermectin inhibits the integrin activation induced by inflammatory cytokines, suggesting that ivermectin is a site 2 antagonist. We showed that TNF, a major pro-inflammatory cytokine, binds to integrin site 2 and induces allosteric integrin activation like other pro-inflammatory cytokines, suggesting that site 2 binding and integrin activation is a potential mechanism of the pro-inflammatory action of these cytokines. Ivermectin suppressed the activation of soluble β3 integrins by TNF and other pro-inflammatory cytokines in a dose-dependent manner in cell-free conditions. Binding to site 2 and the inhibition of binding of inflammatory cytokines may be a potential mechanism of anti-inflammatory action of ivermectin.

## 1. Introduction

Integrins are a superfamily of αβ heterodimers that were originally identified as receptors for extracellular matrix proteins (e.g., fibronectin and laminin), cell surface ligands (e.g., ICAM-1 and VCAM-1), and soluble ligands (e.g., cytokines) [1]. Previous studies identified many new soluble and cell surface ligands using docking simulations with the integrin headpiece as a target. We discovered that several growth factors (e.g., FGF1 [2], FGF2 [3], IGF1 [4], IGF2 [5], and neuregulin-1 [6]) bind to the classical RGD-binding site (site 1). We found that these ligands induce the integrin–cytokine–cognate receptor ternary complex on the cell surface (ternary complex model). Point mutations in the predicted integrin-binding interface in these cytokines suppressed integrin binding and acted as an antagonist of cytokine signaling, although cytokine mutants still bind to cognate receptors [4,5,7]. We propose that growth factor/cytokine binding to site 1 and the subsequent ternary complex formation appear to be critical for their mitogenic action.

In addition to site 1, we discovered that the integrin headpiece has another ligand-binding site (allosteric binding site, site 2), which is on the opposite side of site 1. We discovered that multiple inflammatory cytokines (e.g., CX3CL1, CXCL12, CCL5, CD40L, and CD62P) and pro-inflammatory proteins (e.g., sPLA2-IIA) bind to site 2 and induce integrin activation in cell-free conditions [8,9,10,11,12,13,14] independent of canonical inside-out signaling [15,16]. Hence, we designate this allosteric activation. It is interesting to note that allosteric integrin activation does not require cognitive receptors. A previous study showed that 25-hydroxycholesterol, a major pro-inflammatory lipid mediator, binds to site 2, activates integrins, and induces pro-inflammatory signaling (e.g., IL-6 and TNF secretion) [17]. Therefore, it has been proposed that site 2 is involved in pro-inflammatory signaling. Thus, binding of inflammatory cytokines and inflammatory mediators to site 2 may be a common mechanism of inflammatory signaling. Tumor necrosis factor (TNF) is a major pro-inflammatory cytokine which plays a major role in inflammation, cancer, and insulin resistance [18,19,20]. It is unclear whether TNF binds to integrins or it requires binding to integrins for its signaling functions. It is unclear if TNF binds to site 2 and induces integrin activation and inflammatory signals. Since TNF and CD40L are homologous, we hypothesized that TNF binds to site 2 and induces the allosteric activation of integrins.

Previous studies showed that FGF1 has anti-inflammatory and glucose-reducing activity [21,22,23]. Previous studies also showed that NRG1 has anti-inflammatory and glucose-reducing activity [24,25]. We recently showed that FGF1 and NRG1 bind to site 2 and inhibit integrin activation by pro-inflammatory cytokines instead of activating integrins. These findings suggest that FGF1 and NRG1 compete with pro-inflammatory cytokines for binding to site 2 and thereby inhibit site 2-mediated integrin activation and inflammatory signaling [26,27].

Ivermectin (IVM), a broad-spectrum anthelmintic agent, has anti-inflammatory properties, and affects cellular and humoral immune responses [28]. In addition, IVM has been utilized in a wide variety of conditions, from regulating glucose and cholesterol levels in diabetic mice to suppressing cancer proliferation and inhibition of viral replication. IVM could act through GABA receptors and ligand-gated channels, such as glutamate-gated chloride channels [29]. IVM-mediated inhibition of inflammatory cytokines production may occur through the suppression of the NF-κB pathway [30]. It has also been reported that ivermectin is a high-affinity ligand for the farnesoid X receptor (FXR), which has important roles in maintaining bile acid and cholesterol homeostasis. [31]. Also, previous studies showed that IVM binds to the receptor-binding domain (RBD) of the spike protein of SARS-CoV-2 [32]. The mechanism of action of IVM, however, has not been established. Since IVM has anti-inflammatory action, like FGF1 and NRG1, we hypothesized that IVM may bind to site 2 and inhibit the binding of pro-inflammatory cytokines to site 2 and thereby suppress inflammatory signaling.

In the present study, we first performed a docking simulation of the interaction between IVM and site 2. The simulation predicted that IVM binds to site 2 and that its binding site overlapped with those of pro-inflammatory cytokines. Importantly, IVM inhibited the allosteric activation of integrins by TNF and several inflammatory cytokines. IVM binding to site 2 may be a mechanism of the anti-inflammatory action of IVM. We also performed a docking simulation between TNF and site 2. The simulation predicts that TNF binds to site 2.

We showed that TNF binding to site 2 induced the allosteric activation of integrins, like several other inflammatory cytokines, suggesting that allosteric integrin activation may be a potential marker of pro-inflammatory signaling and insulin resistance, and site 2 may be a potential therapeutic target.

## 2. Results

### 2.1. Docking Simulation Predicts That IVM Binds to Site 2 of Integrin αvβ3

Previous studies showed that the anti-inflammatory cytokines FGF1 and NRG1 (site 2 antagonists) inhibit the binding of pro-inflammatory cytokines (site 2 agonists) and subsequent integrin activation (see Introduction). FGF1 and NRG1 are also known to have glucose-reducing activity (see introduction). Previous studies suggest that the anti-parasite agent IVM has potential anti-inflammatory action, and hypoglycemia is a serious side-effect of ivermectin [33]. Therefore, we hypothesized that ivermectin may bind to site 2 and inhibit the binding of pro-inflammatory cytokines such as TNF. To address this hypothesis, we performed a docking simulation of the interaction between integrin αvβ3 (closed headpiece, 1JV2.pdb) and IVM. The simulation predicted that several docking poses are in the first cluster (docking energy—18.24 kcal/mol) (Figure 1a), predicting that they represent poses when IVM binds to site 2 (Figure 1b). Predicted amino acid residues involved in IVM binding are shown in Table 1 and Figure 1c. The predicted IVM-binding site is in the center of site 2 and overlaps with those of multiple pro-inflammatory cytokines (Figure 2), suggesting that IVM may inhibit integrin activation by pro-inflammatory cytokines.

### 2.2. IVM Inhibits Allosteric Integrin Activation Induced by Multiple Pro-Inflammatory Cytokines

We studied if IVM inhibits the binding of multiple pro-inflammatory cytokines to site 2. We measured the effect of IVM on the activation of soluble integrins by pro-inflammatory cytokines in cell-free conditions. IVM suppressed the activation of αvβ3 (Figure 3a–c) and αIIbβ3 (Figure 4a–c) by FGF2, CD40L, and CCL5, suggesting that IVM acts as an antagonist for site 2 in αvβ3 and αIIbβ3. This is consistent with the idea that the anti-inflammatory action of IVM is mediated by blocking the binding of multiple inflammatory cytokines to site 2.

### 2.3. Docking Simulation Predicts That TNF Binds to Integrin

Previous studies showed that CD40L, which is a member of the TNF family, binds to site 1 of integrin αvβ3 through the trimeric interface, suggesting that integrin-bound CD40L is a monomer [34]. CD40L binds to site 2 of integrin αvβ3 and allosterically activates integrins [8,9], suggesting that this is a potential mechanism of pro-inflammatory action of CD40L. TNF is a one of the most potent pro-inflammatory cytokines, and homologous to CD40L. Previous studies suggest that pro-inflammatory cytokines (e.g., TNF) induce inflammatory signaling and insulin resistance [35,36], suggesting that allosteric activation may be a potential marker of inflammatory signaling and insulin resistance. We thus studied if TNF binds to integrins. We hypothesized that TNF binds to site 2 and activates integrins.

A docking simulation predicted that TNF binds to site 2 of integrins (docking energy—19.15 kcal/mol) and activates soluble integrins (Figure 5). Several docking poses clustered well in the first cluster (Figure 5b). Predicted amino acid residues of αvβ3 for binding to TNF are shown in Table 2.

The site 2-binding interface is present on the outer-surface of the TNF trimer as in CD40L (Figure 5a), suggesting that trimeric TNF binds to site 2 and activates integrins as in CD40L.

We studied whether TNF activates soluble integrins in ELISA-type activation assays. We found that TNF activated soluble integrins αvβ3 and αIIbβ3 in a dose-dependent manner in cell-free conditions (Figure 5c,d). These findings suggest that TNF, like several other pro-inflammatory cytokines, binds to site 2 and activates integrins in a dose-dependent manner. It should be noted that TNF-induced activation of soluble integrins is independent of TNF receptors or inside-out signaling, like integrin activation induced by several other pro-inflammatory cytokines (see Introduction).

We found that IVM inhibited TNF-induced integrin activation to the level comparable to that of NRG1 (Figure 6b,c). These findings are consistent with the idea that NRG1 and IVM inhibit site 2-mediated integrin activation as site 2 antagonists. IVM is known to be insoluble in water (max. 5 μg/mL), and we used stock IVM (10 mM) in 95% ethanol and diluted in water. Although we showed that calculated 100 μM IVM effectively inhibited the allosteric activation of integrins, the actual IVM concentration may not be higher than 5 μg/mL (approx. 5 μM). If the binding of IVM to site 2 is critical for its anti-inflammatory action, it would be possible to design more effective or water-soluble IVM variants.

## 3. Discussion

IVM has been widely used clinically, originally as an anti-parasite agent. It has been proposed that IVM is potentially an anti-inflammatory, anti-cancer and anti-virus agent as well. However, it has not been systematically tested clinically if IVM is effective as an anti-inflammatory agent. FGF1 and NRG1, which are anti-inflammatory and have glucose-lowering action, bind to site 2 and block integrin activation by pro-inflammatory cytokines (see Section 1). The present study provides evidence for the first time that IVM is an inhibitor of site 2-mediated integrin activation (and the subsequent inflammatory signaling) (Figure 7). In this model, multiple cytokines that induce inflammatory signals and glucose-lowering action bind to site 2, suggesting that site 2-medicated integrin activation is a potential marker of pro-inflammatory action and insulin resistance. In contrast NRG1, FGF1 and IVM bind to site 2, but act as inhibitors of site 2-mediated pro-inflammatory action and insulin resistance (see Section 1). However, the present study did not provide information on how the site 2 binding is involved in inflammation. One possibility is that ligand binding to site 2 induces outside-in signaling and affects inflammatory signaling inside the cells as previously suggested [17].

Also, inflammation is involved in cancer proliferation and multiple pro-inflammatory cytokines (e.g., CCL5, CX3CL1, and CXCL12) play a role in cancer. Blocking pro-inflammatory signals through site 2 may suppress cancer proliferation. Activation of platelet integrin αIIbβ3 is a key event in platelet activation and thrombus formation [15,37]. We previously showed that multiple cytokines bind to site 2 of αIIbβ3 and activate this integrin [13]. It is highly likely that multiple inflammatory cytokines allosterically activate αIIbβ3 and induce platelet activation and thrombosis [28,30,32]. We showed that IVM inhibited the allosteric activation of αIIbβ3 by inflammatory cytokines. IVM may thus have anti-thrombotic action (e.g., deep vein thrombosis). The present study is expected to facilitate the elucidation of the mechanism of action of IVM and repurposing of IVM.

Notably, we showed that TNF, a major pro-inflammatory cytokine, binds to site 2, and induced the allosteric activation of integrin αvβ3 and αIIbβ3, like several other pro-inflammatory cytokines. It has been reported that CD40L binds to integrins and CD40 simultaneously [38]. In the present study, docking simulations predict that a TNF monomer may bind to integrin and TNFR. It will be interesting to study if TNF induces the integrin–TNF–TNFR ternary complex on the cell surface in future studies.

The present study was started as prediction by docking simulation, and the prediction was tested in cell-free conditions. We will confirm the observations in cell-based assays in the future experiments.

## 4. Materials and Methods

### 4.1. Materials

Ivermectin (IVM) was obtained from eMolecules (San Diego, CA, USA). We used stock IVM (10 mM) in 95% ethanol and diluted it in water. Human soluble αvβ3 (IT3-H52E3) and αIIbβ3 (IT3-H52W8) were obtained from ThermoFischer (Waltham, MA, USA). Mab AV10 (to human β3) was kindly provided by Brunie Felding (Scripps Research Institute, La Jolla, CA, USA). HRP-conjugated anti-His tag (C-terminal) antibody was purchased from Qiagen (Valencia, CA, USA).

### 4.2. Protein Expression

The truncated fibrinogen γ-chain C-terminal domain (γC399tr) was generated as previously described [39]. Fibrinogen γ-chain C-terminal residues 390–411 cDNA encoding (6 His-tagged) [HHHHHH]NRLTIGEGQQHHLGGAKQAGDV] was conjugated with the C-terminus of GST (designated γC390–411) in a pGEXT2 vector (BamHI/EcoRI site). The protein was synthesized in *E. coli* BL21 and purified using glutathione affinity chromatography. FGF2 was synthesized as previously described [3]. CCL5 [13], CXCL12 [12], CX3CL1 [40] were synthesized as described. TNF was synthesized as described for CCL5 and CX3CL1. Briefly, cDNA for TNF [VRSSSRTPSDKPVAHVVANPQAEGQLQWLNRRANALLANGVELRDNQLVVPSEGLYLIYSQVLFKGQGCPSTHVLLTHTISRIAVSYQTKVNLLSAIKSPCQRETPEGAEAKPWYEPIYLGGVFQLEKGDRLSAEINRPDYLDFAESGQVYFGIIAL] was synthesized and subcloned into the BamH1/EcoRI site of PET28a vector. The protein was synthesized in *E. coli* BL21 (endotoxin-free, Clear coli) as insoluble inclusion bodies. TNF was purified in denaturing conditions in 8 M urea in Ni-NTA affinity chromatography and refolded as described [40].

### 4.3. Docking Simulation

A docking simulation of the interaction between TNF/IVM and integrin αvβ3 (closed head-piece form, PDB code 1JV2) was performed using AutoDock3, as described previously [6]. We used the headpiece (residues 1–438 of αv and residues 55–432 of β3) of αvβ3 (closed form, 1JV2.pdb). Cations were not present in integrins during the docking simulation. In the AutoDock 3.05 program, the ligand is presently compiled to a maximum size of 1024 atoms. The solvent-exposed Mg2 octahedral vertex was left empty in the model during docking calculations. Atomic solvation parameters and fractional volumes were assigned to the protein atoms by using the AddSol utility (Autodock3.05), and grid maps were calculated by using the AutoGrid utility in AutoDock3.05. A gridmap with 127 127 127 points and a grid point spacing of 0.603 Å included the site 2 in the I-like domain of b3, which are large enough to accommodate the TNF or IVM structure. Kollman “united-atom” charges were used. AutoDock 3.05 uses a Lamarckian genetic algorithm (LGA) that couples a typical Darwinian genetic algorithm for global searching with the Solis and Wets algorithm for local searching. The LGA parameters were defined as follows: the initial population of random individuals had a size of 50 individuals; each docking was terminated with a maximum number of 1 × 10^6^ energy evaluations or a maximum number of 27,000 generations, whichever came first; mutation and crossover rates were set at 0.02 and 0.80, respectively. An elitism value of 1 was applied, which ensured that the top ranked individual in the population always survived into the next generation. A maximum of 300 iterations per local search was used. The probability of performing a local search on an individual was 0.06, whereas the maximum number of consecutive successes or failures before doubling or halving the search step size was 4. This set of parameters was used for all dockings.

### 4.4. Binding of Soluble Integrins to TNF

ELISA-type binding assays were performed as described previously [41]. Briefly, wells of 96-well Immulon 2 microtiter plates (Dynatech Laboratories, Chantilly, VA) were coated with 100 mL PBS containing TNF for 2 h at 37 °C. The remaining protein-binding sites were blocked by incubating with PBS/0.1% BSA for 30 min at room temperature. After washing with PBS, the soluble recombinant αvβ3 or αIIbβ3 (1 μg/mL) was added to the wells and incubated in HEPES-Tyrodes buffer (10 mM HEPES, 150 mM NaCl, 12 mM NaHCO_3_, 0.4 mM NaH_2_PO_4_, 2.5 mM KCl, 0.1% glucose, 0.1% BSA) with 1 mM MnCl_2_ for 1 h at room temperature. After unbound integrin was removed by rinsing the wells with binding buffer, bound integrin was measured using anti-integrin β3 mAb (AV-10) followed by HRP-conjugated goat anti-mouse IgG and peroxidase substrates. Tecan SPARK ELISA reader at 450 nm was used. Data were analyzed using Prism 10.

### 4.5. Activation of Soluble αIIbβ3 and αvβ3

ELISA-type integrin activation assays were performed as described previously [11,40]. Briefly, wells of 96-well Immulon 2 microtiter plates (Dynatech Laboratories, Chantilly, VA, USA) were coated with 100 µL 0.1 M PBS containing γC390-411 for αIIbβ3 and γC399tr for αvβ3 for 2 h at 37 °C. The remaining protein-binding sites were blocked by incubating with PBS/0.1% BSA for 30 min at room temperature. After washing with PBS, soluble recombinant αIIbβ3 or αvβ3 (1 µg/mL) in the presence or absence of FGF1 and/or FGF2 was added to the wells and incubated in Hepes–Tyrodes buffer (10 mM HEPES, 150 mM NaCl, 12 mM NaHCO_3_, 0.4 mM NaH_2_PO_4_, 2.5 mM KCl, 0.1% glucose, 0.1% BSA) with 1 mM CaCl_2_ for 1 h at room temperature. After unbound αIIbβ3 or αvβ3 was removed by rinsing the wells with binding buffer, bound αIIbβ3 or αvβ3 were measured using anti-integrin β3 mAb (AV-10) followed by HRP-conjugated goat anti-mouse IgG and peroxidase substrates.

### 4.6. Statistical Analysis

Treatment differences were tested using ANOVA and Tukey multiple comparison tests to control the global type I error using Prism 10 (Graphpad Software, Boston, MA, USA).

## 5. Conclusions

We investigated the ivermectin (IVM) interaction with allosteric site 2 of integrins and its effect on the pro-inflammatory signaling associated with this site. By docking simulation studies, we have shown that IVM is predicted to be a site 2 antagonist, and TNF was predicted to be its agonist and to allosterically activate integrins, which could also be a potential mechanism of the pro-inflammatory action of TNF. In cell-free conditions, we demonstrated an inhibitory effect of IVM on the activation of β3 integrins in the presence of TNF, FGF2, CD40L, and CCL5 in a dose-dependent manner. This points at a possible mechanism of anti-inflammatory action of ivermectin.

## Figures and Tables

**Figure 1 ijms-26-08655-f001:**
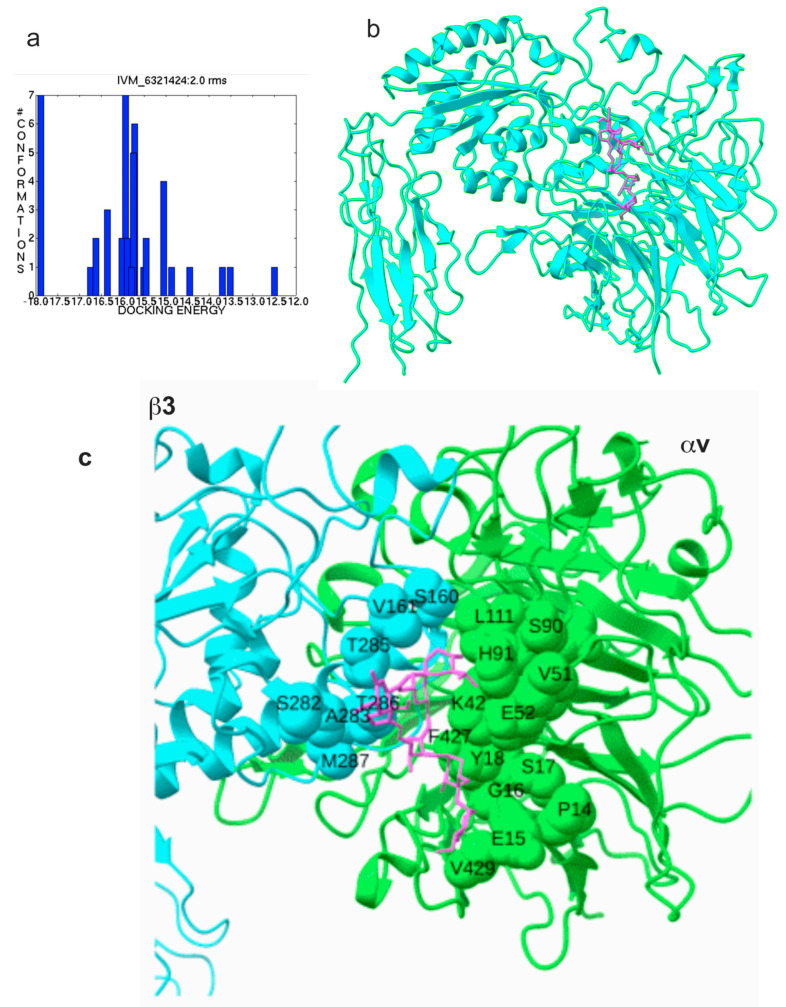
Docking simulation of interaction between integrin αvβ3 and ivermectin (IVM). (**a**) Clustering of docking poses. Autodock tools were used. (**b**) Docking simulation between αvβ3 (closed headpiece, 1JV2.pdb) and IVM was performed using autodock3. Docking energy is −18.24 kcal/mol. Pose in the cluster 1 is shown. (**c**) Amino acid residues of αvβ3 surrounding IVM in site 2.

**Figure 2 ijms-26-08655-f002:**
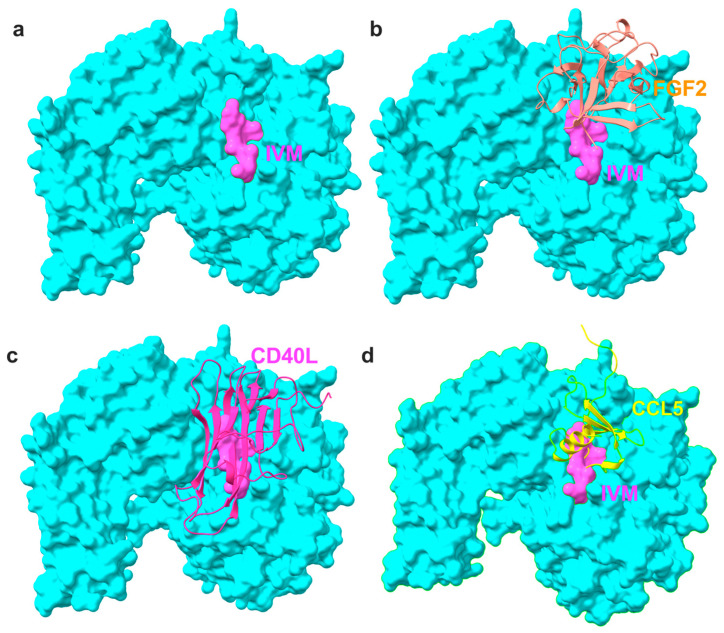
Positions of IVM and inflammatory cytokines on site 2 of αvβ3. IVM and multiple inflammatory cytokines that bind to site 2 were superposed. Docking poses of inflammatory cytokines were taken from previous papers. (**a**) IVM only, (**b**) FGF2 [26] and IVM, (**c**) CD40L [9] and IVM, and (**d**) CCL5 [13] and IVM.

**Figure 3 ijms-26-08655-f003:**
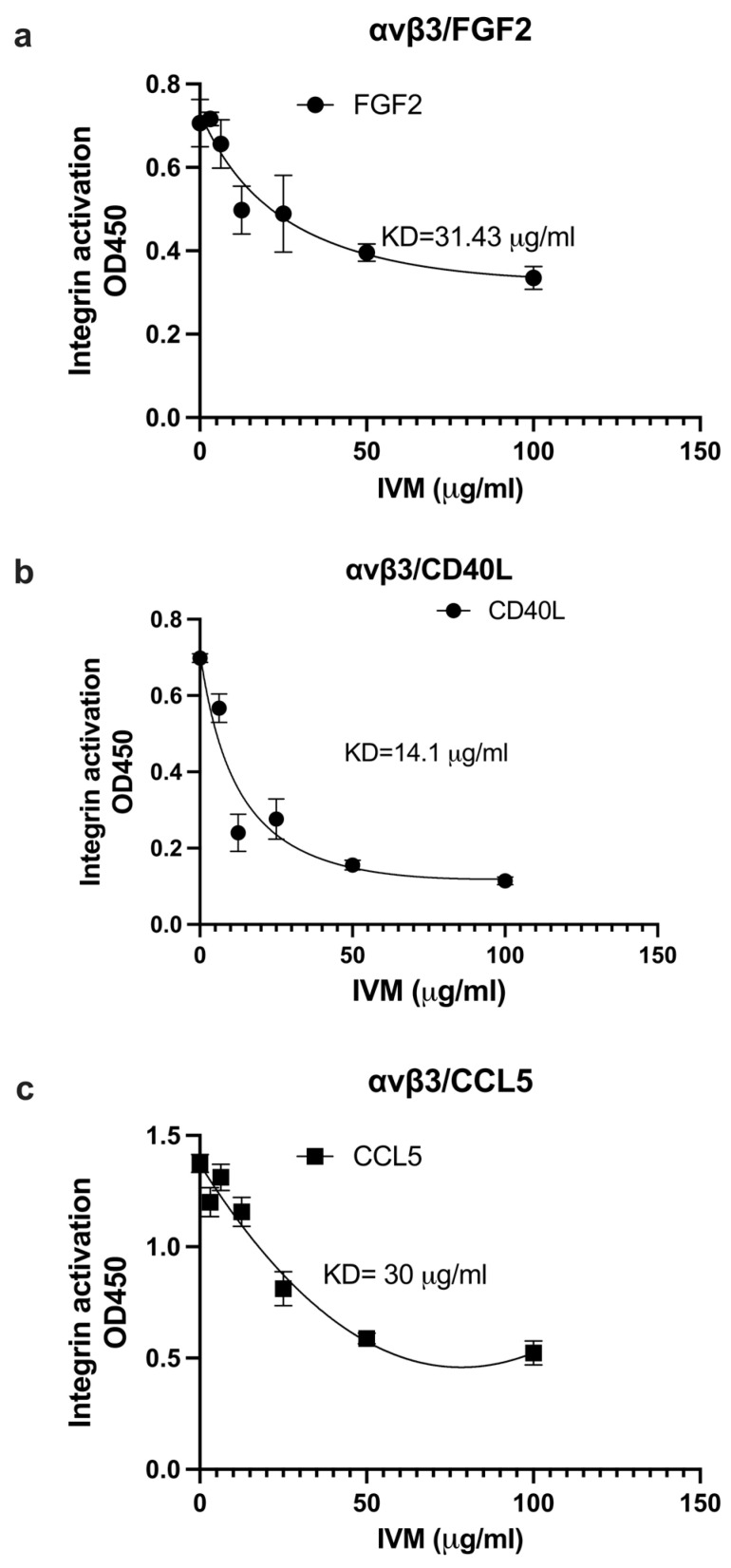
Inhibition by IVM of pro-inflammatory cytokine-mediated activation of integrin αvβ3. Wells of 96-well microtiter plate were coated with the C-terminal domain of fibrinogen γ-chain in which the C-terminal 12 residues are truncated (γC399tr), a specific ligand for integrin αvβ3 (50 μg/mL). Soluble αvβ3 was activated by (**a**) FGF2 (12.5 μg/mL), (**b**) CD40L (25 μg/mL), (**c**) CCL5 (at 6.25 μg/mL) in the presence of IVM. Data is shown as means +/− SD in triplicate experiments.

**Figure 4 ijms-26-08655-f004:**
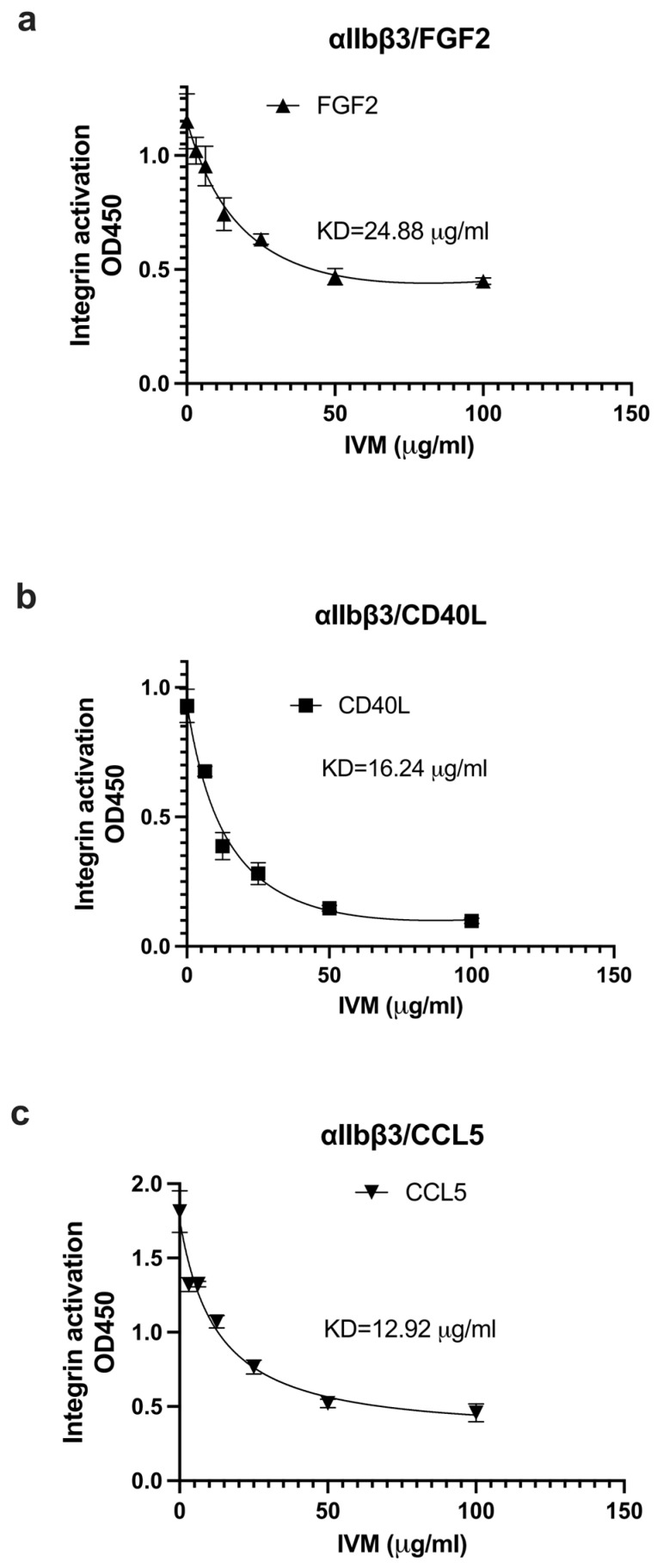
Inhibition by IVM of pro-inflammatory cytokine-mediated activation of integrin αIIbvβ3. Wells of 96-well microtiter plate were coated with the C-terminal domain of fibrinogen γ-chain C-terminal 12 residues (γC390-411), a specific ligand for integrin αIIbβ3 (20 μg/mL). Soluble αIIbβ3 (1 μg/mL) was activated by (**a**) FGF2 (12.5 μg/mL), (**b**) CD40L (25 μg/mL), (**c**) CCL5 (at 6.25 μg/mL) in the presence of IVM. Data is shown as means +/− SD in triplicate experiments.

**Figure 5 ijms-26-08655-f005:**
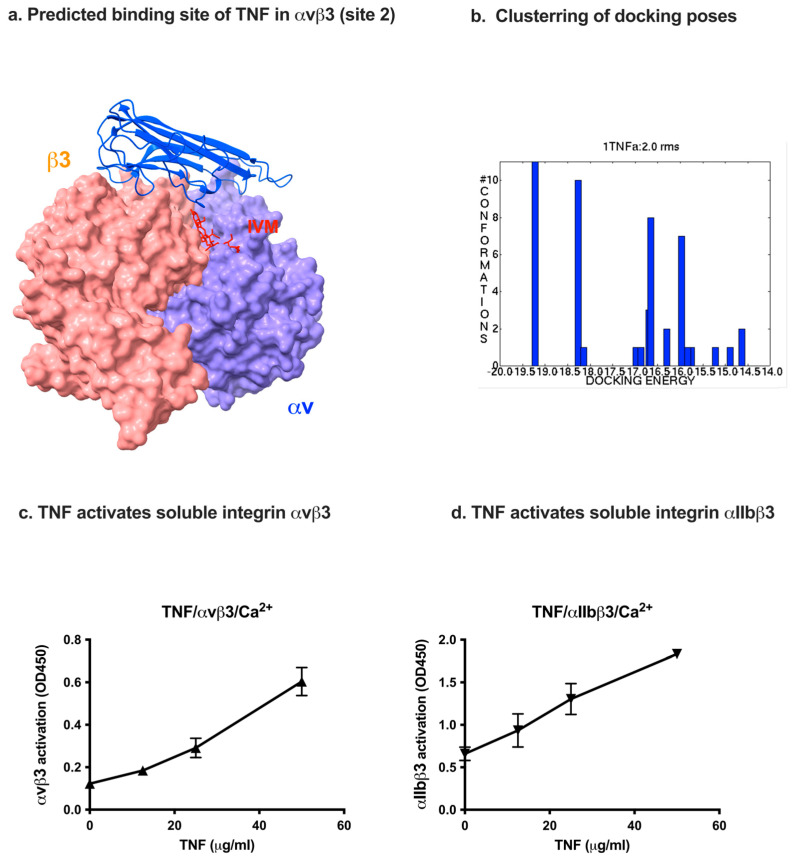
Binding of TNF to site 2 of αvβ3 allosterically activates soluble integrins (**a**). Docking simulation between αvβ3 (closed headpiece, 1JV2.pdb) and TNF was performed using autodock3. Position of IVM (red) (see Figure 1) and TNF (blue) are shown. IVM and TNF-binding sites overlap (**b**). Docking pose of IVM was superposed. The pose in the first cluster is used for further analysis (**c**,**d**). TNF activates integrins αvβ3 (**c**) and αIIbβ3 (**d**). Wells of 96-well Immulon 2 microtiter plates were coated with 100 μL PBS containing γC390-411 for αIIbβ3 and γC399tr for αvβ3 for 2 h at 37 °C. Soluble recombinant αIIbβ3 or αvβ3 (1 μg/mL) in the presence of TNF was added to the wells and incubated in Hepes–Tyrodes buffer with 1 mM CaCl_2_ for 1 h at room temperature. Bound αIIbβ3 or αvβ3 was measured using anti-integrin β3 mAb (AV-10) followed by HRP-conjugated goat anti-mouse IgG and peroxidase substrates.

**Figure 6 ijms-26-08655-f006:**
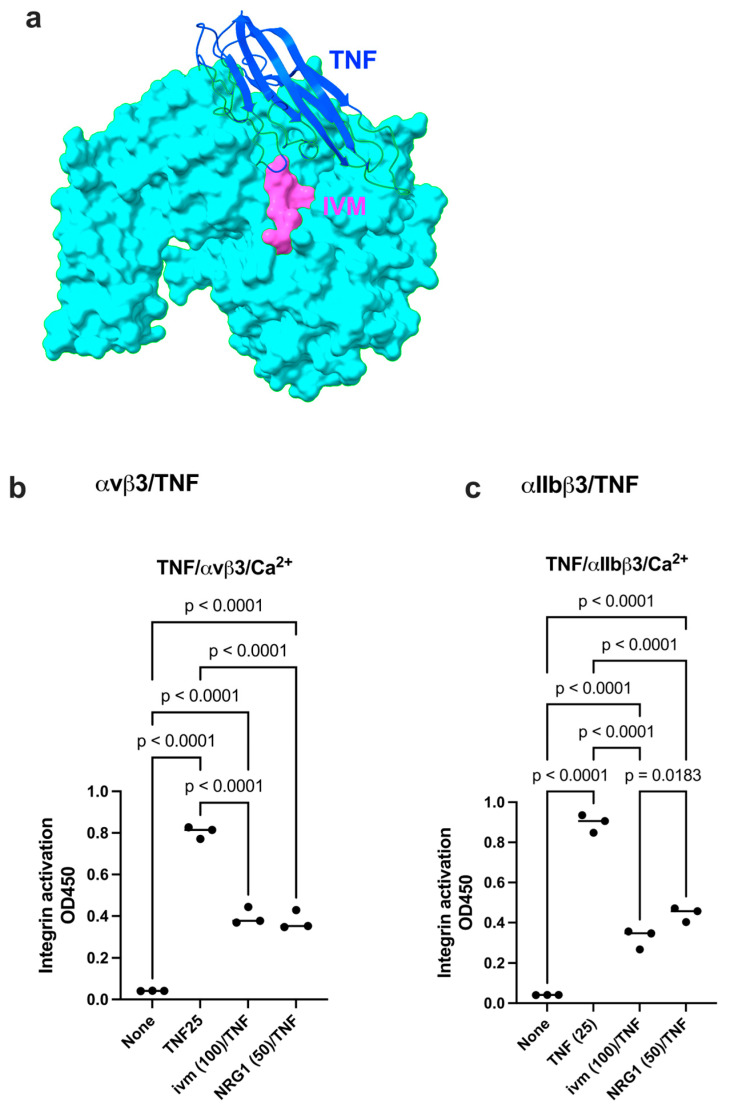
Inhibition by IVM or NRG1 of TNF-mediated activation of integrin αvβ3 and αIIbβ3. (**a**) Positions of NRG1 and IVM. (**b**,**c**) Wells of 96-well microtiter plate were coated with the C-terminal domain of fibrinogen γ-chain C-terminal 12 residues (γC390-411), a specific ligand for integrin αIIbβ3 (20 μg/mL). Soluble αIIbβ3 (1 μg/mL) was activated by TNF (25 μg/mL) in the presence of IVM (100 μg/mL), or NRG1 (50 μg/mL). Data is shown as means +/− SD in triplicate experiments.

**Figure 7 ijms-26-08655-f007:**
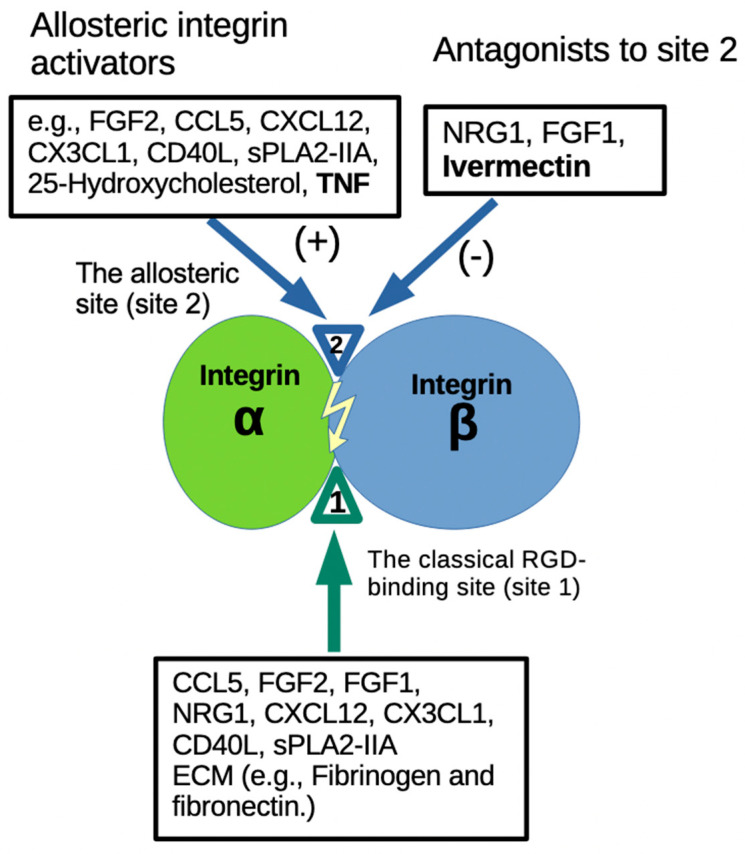
A model of the inhibition of the binding of pro-inflammatory cytokines to the allosteric site 2 by anti-inflammatory cytokines and ivermectin. Previous studies showed that multiple pro-inflammatory cytokines and 25-hydoxycholesterol bind to the center of site 2 and induce integrin activation and inflammatory signals (see Section 1). TNF also binds to site 2 and induces integrin activation (the present study). In contrast, FGF1 and NRG1 bind to site 2 (off the center of site 2) and inhibit integrin activation by inflammatory cytokines (see Section 1). Ivermectin binds to the center of site 2 and inhibits the binding of pro-inflammatory cytokines and the resulting integrin activation and inflammatory signals. These findings suggest that ivermectin binding to site 2 is a potential mechanism of anti-inflammatory action of ivermectin.

**Table 1 ijms-26-08655-t001:** Predicted amino acid residues involved in binding to ivermectin in integrin αvβ3.

αv (1JV2.pdb)	β3 (1JV2.pdb)
Pro14, Glu15, Gly16, Ser17, Tyr18, Lys42, Val51, Glu52, Ser90, His91, Gln92, Trp93, Leu111, Phe427, Gly428, Val429, Asp430	Val161, Ser162, Ser282, Ala283, Thr285, Thr286, Met287

Amino acid residues within 0.6 nm between ivermectin and αvβ3 were selected using Pdb Viewer (version 4.1).

**Table 2 ijms-26-08655-t002:** Predicted amino acid residues involved in binding of TNF to site 2 in integrin αvβ3.

TNF (1TNF.pdb)	αv (1JV2)	β3 (1JV2)
Asn19, Glu23, Gln25, Gln27, Trp28, leu29, Arg31, Glu42, Arg44, Asp45, Asn46, Gln47, Val49, Cys69, Pro70, Ser71, Thr72, His73, Leu75, Leu76, Thr77, Thr79, Ser81, Ile83, Val85, Gln88, Thr89, Lys90, Val91, Asn92, Ile97, Gln102, Arg103, Thr105, Glu107, Gly129, Arg131, Glu135, Asn137	Gly76, Asn77, Asp79, Ala81, Lys82, Asp83, Asp84, Pro85	Glu171, Glu174, Asn175, Thr182, Thr183, Cys184, Leu185, Pro186, Lys191, His192, Val193, Leu194, Thr195, Arg202, Glu205, Glu206, Val207, Lys208, Lys209, Gln210, Ser211, Gly276, Ser277, Asp278 Asn279, His280, Ser282, Thr285, Thr286

Amino acid residues within 0.6 nm between ivermectin and αvβ3 were selected using Pdb Viewer (version 4.1).

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
