# Peer review of "Ivermectin Binds to the Allosteric Site (Site 2) and Inhibits Allosteric Integrin Activation by TNF and Other Pro-Inflammatory Cytokines"

_ijms, 2025, doi:10.3390/ijms26178655_

Round 1
Reviewer 1 Report
Comments and Suggestions for Authors
This research showed that ivermectin suppressed the activation of soluble β3 integrins by TNF and other pro-inflammatory cytokines. It is interesting. I have the following comments.
Not all chemokines are pro-inflammatory cytokines. Authors need to distinguish them.
Figures need to be described in detail. For example, “IVM suppressed activation αvβ3 (Figure 3a-d) and αIIbβ3 (Figure 4 a-d) 116 by FGF2, CD40L, and CCL5”. In addition, I did not see Figure 3d ad Figure 4d.
Base on “Previous studies showed that CD40L, which is a member of the TNF family”, authors focus on TNF. As for FGF2 and CCL5, what do they bind to?
Author Response
Comment #1. Not all chemokines are pro-inflammatory cytokines. Authors need to distinguish them.
Response#1. CX3CL1, CXCL12, CCL5, CD40L, CD62P, FGF2, and TNF are all pro-inflammatory. FGF1 and NRG1 are anti-inflammatory. We described this in the introduction in the original version.
Comment #2. Figures need to be described in detail. For example, “IVM suppressed activation αvβ3 (Figure 3a-d) and αIIbβ3 (Figure 4 a-d) 116 by FGF2, CD40L, and CCL5”. In addition, I did not see Figure 3d ad Figure 4d.
Response #2. Figure 3d and Figure 4d were removed.
Comments #3. Base on “Previous studies showed that CD40L, which is a member of the TNF family”, authors focus on TNF. As for FGF2 and CCL5, what do they bind to?
Response #3. FGF2 and CCL5 bind to site 2 and allosterically activate integrins. We included references in the introduction.
Reviewer 2 Report
Comments and Suggestions for Authors
The paper by Takada and Takada is dedicated to investigation of ivermectin (IVM) interaction with allosteric site 2 of integrins and its effect on the proinflammatory signaling associated with this site. By docking simulation studies, the authors have shown that IVM is predicted to be site 2 antagonist, meanwhile, TNFa was predicted to be its agonist and to allosterically activate integrins, which could be also a potential mechanism of pro-inflammatory action of TNFa. In cell free conditions the authors demonstrated inhibitory effect of IVM on the activation of beta3 itegrins in the presence of TNFa, FGF2, CD40L, and CCL5 in a dose-dependent manner. This points at possible mechanism of anti-inflammatory action of ivermectin.
Here are my comments to be considered by the authors.
General
- Please carefully check for the formatting throughout the article: Fig.4, fonts (lines 376-386), references.
- Please check the grammar throughout the paper (the correct use of tenses etc. )
- Please avoid repeats of the same phrases (lines 19-20, lines 59-60 - “it is unclear if TNF binds…”, lines 63-65 – the sentences are almost completely, line 115 –“binding”, line 350 etc.). It seems that these sentences were written with the help of AI, please, avoid this.
Particular
Title
I would recommend not to mention «a potential mechanism of anti-inflammatory action of IVM» in the title, and leave it for the Discussion and Conclusion parts.
Abstract
Lines 20-21 – «Ivermectin binds to site 2 and inhibits the integrin activation induced by inflammatory cytokines» - it sounds like it has been demonstrated that IVM binds site 2, but it has only been shown by docking simulation studies. Please correct the sentence.
Introduction
- Please add the information upon the role of integrins activation in the inflammation.
- Line 80 –«The mechanism of of IVM» - a missing word here.
Results
- The section «IVM inhibits allosteric integrin activation induced by multiple pro-inflammatory cytokines» - Please add the short description of the method used to obtain these results.
- Line 220 – wrong links to Figures.
- 6a – is it TNFa or NRG1 indicated in blue?
- Lines 225-227 and Figure 6 – Please introduce NRG1 and its relation to the experiments as it is unclear from previous part of the Results. Also, please describe the obtained results in more details, with the links to the relevant figures.
Discussion
- Please rewrite the Discussion to make the text more logical and coherent.
- Lines 271-273 – The sentence seems to be unrelated to the rest of the text.
- Lines 273-274 – «It has been proposed that IVM is potentially anti-inflammatory, anti-cancer and anti-virus» - a missed noun.
- The Discussion misses references, e.g. lines 273-274, 296 etc.
- Lines 312-317 - This information seems to be not appropriate for the Discussion. It should be reflected in the Results and a short summary can be then given in the Discussion.
- I would recommend to add the comment that as the results of this study were obtained in cell-free conditions, to continue working in this direction, they should be confirmed in cell models. Otherwise, they should be interpreted with care.
- Please add the concluding sentences – either in Discussion or as a separate Conclustion part.
Methods
- Lines 376-386 – seem to repeat the lines 366-375, please, shorten the text.
- Please add the details on the ELISA experiments (the machine, the calculations).
- Please add the information on how the assays with IVM were performed, including the information on how it was dissolved.
Comments on the Quality of English Language
Please check the grammar throughout the paper (the correct use of tenses etc. ).
Author Response
Comment #1
- Please carefully check for the formatting throughout the article: Fig.4, fonts (lines 376-386), references.
- Please check the grammar throughout the paper (the correct use of tenses etc. )
- Please avoid repeats of the same phrases (lines 19-20, lines 59-60 - “it is unclear if TNF binds…”, lines 63-65 – the sentences are almost completely, line 115 –“binding”, line 350 etc.). It seems that these sentences were written with the help of AI, please, avoid this.
Response #1. We corrected the errors.
Comment #2.
I would recommend not to mention «a potential mechanism of anti-inflammatory action of IVM» in the title, and leave it for the Discussion and Conclusion parts.
Response #2. «a potential mechanism of anti-inflammatory action of IVM» was removed from the title and discussed in Discussion section.
Comment #3. Lines 20-21 – «Ivermectin binds to site 2 and inhibits the integrin activation induced by inflammatory cytokines» - it sounds like it has been demonstrated that IVM binds site 2, but it has only been shown by docking simulation studies. Please correct the sentence.
Response #3. Sentence was changed to IVM binding to site 2 is a prediction.
Comments #4. Please add the information upon the role of integrins activation in the inflammation
Response #4. Previous studies showed that 25-hydroxycholesterol, a pro-inflammatory lipid mediator, binds to site 2 and this leads to secretion of pro-inflammatory cytokines (IL-6 and TNF) through NF-kB activation (in Introduction). Integrin activation induces FAK and MAPK activation. But it is unclear how integrin activation plays a role in inflammation.
Comments #5. Line 80 –«The mechanism of of IVM» - a missing word here.
Response #5. We changed to "The mechanism of IVM action".
Comment #6. The section «IVM inhibits allosteric integrin activation induced by multiple pro-inflammatory cytokines» - Please add the short description of the method used to obtain these results.
Response #6. Short description of the method used was included.
Comment #7
Line 220 – wrong links to Figures.
Response #7 Correct links were included.
Comment #8 6a – is it TNFa or NRG1 indicated in blue?
Response #8. It is TNF.
Comment #9. Lines 225-227 and Figure 6 – Please introduce NRG1 and its relation to the experiments as it is unclear from previous part of the Results. Also, please describe the obtained results in more details, with the links to the relevant figures.
Response #9. NRG1 is known to be anti-inflammatory and we recently predicted that NRG1 binds to site 2 (We introduced NRG1 in Introduction).
Comment #10. Please rewrite the Discussion to make the text more logical and coherent.
Response #10. The discussion section was edited and reorganized for clarity.
Comment #11. Lines 271-273 – The sentence seems to be unrelated to the rest of the text.
Response #11. The section was reorganized. "We showed how Pro-inflammatory cytokines, anti-inflammatory cytokines, and ivermectin interact with site 2 (Figure 7).
Comment #12. Lines 273-274 – «It has been proposed that IVM is potentially anti-inflammatory, anti-cancer and anti-virus» - a missed noun.
Response #12. We changed the sentence to "It has been proposed that IVM is potentially anti-inflammatory, anti-cancer and anti-virus "agent" as well. "
Comment #13.
The Discussion misses references, e.g. lines 273-274, 296 etc.
Lines 312-317 - This information seems to be not appropriate for the Discussion. It should be reflected in the Results and a short summary can be then given in the Discussion.
I would recommend to add the comment that as the results of this study were obtained in cell-free conditions, to continue working in this direction, they should be confirmed in cell models. Otherwise, they should be interpreted with care.
Please add the concluding sentences – either in Discussion or as a separate Conclustion part.
Response #13. Several references were included. Part of discussion was moved to the Result section.
We added the following statement
"The present study was started as prediction by docking simulation, and the prediction was tested in cell-free conditions. We will confirm the observations in cell-based assays in the future experiments."
"Conclusion" was added.
Comment #14. Lines 376-386 – seem to repeat the lines 366-375, please, shorten the text.
Response #14. We did not find the repeat, unfortunately. Comment #15
- Please add the details on the ELISA experiments (the machine, the calculations).
- Please add the information on how the assays with IVM were performed, including the information on how it was dissolved.
Response #15.
More detailed description of the ELISA measurement.
We used stock IVM (10 mM) in 95% ethanol and diluted in water. We added this statement in the method section.
Round 2
Reviewer 1 Report
Comments and Suggestions for Authors
I have no comments.